# Immunotherapy Enhancement by Targeting Extracellular Tumor pH in Triple-Negative Breast Cancer Mouse Model

**DOI:** 10.3390/cancers15204931

**Published:** 2023-10-11

**Authors:** Azizur Rahman, Branislava Janic, Tasnim Rahman, Harshit Singh, Haythem Ali, Ramandeep Rattan, Mohsin Kazi, Meser M. Ali

**Affiliations:** 1Department of Neurosurgery, Henry Ford Hospital, Detroit, MI 48202, USA; 2Department of Radiation Oncology, Henry Ford Hospital, Detroit, MI 48202, USA; 3Women’s Health Services, Henry Ford Hospital, Detroit, MI 48202, USArrattan1@hfhs.org (R.R.); 4Department of Pharmaceutics, College of Pharmacy, King Saud University, Riyadh 11451, Saudi Arabia; mkazi@ksu.edu.sa

**Keywords:** TNBC, extracellular pH (pH_e_), sodium bicarbonate (NaHCO_3_), PD-L1, immunotherapy

## Abstract

**Simple Summary:**

Despite remarkable progress in immunotherapy, most advanced cancer patients still show intrinsic or naturally acquired resistance to immune checkpoint blockade strategies leading to treatment failure. Breast cancer produces lactate and protons in tumor cells, which are secreted to maintain intracellular homeostasis, causing extracellular pH (pH_e_) of the tumor microenvironment to become acidic. Immune cells are unfortunately unable to mount effective responses under these harsh conditions—thereby causing immunotherapy treatments to fail. Therefore, reducing tumor acidity can be effective in cancer immunotherapy. The role of neutralizing tumor acidosis in improving treatment response in triple negative breast cancer (TNBC) environment is not known and is thus the focus of our study. In this study, oral administration of either sodium bicarbonate or sodium bicarbonate plus anti-PD-L1 combination enhanced responses to anti-tumor immunity by tumor growth inhibition and improving survival time in TNBC. Our key finding is a pH-neutralizer for improving cancer immunotherapy. The combination of these therapies can provide exceptional impact by providing powerful personalized medicine paradigm.

**Abstract:**

Triple-negative breast cancer (TNBC), as one of the most aggressive forms of breast cancer, is characterized by a poor prognosis and a very low rate of disease-free and overall survival. In recent years, immunotherapeutic approaches targeting T cell checkpoint molecules, such as cytotoxic lymphocyte antigen-4 (CTLA-4), programmed death1 (PD-1) or its ligand, programmed death ligand 1 (PD-L1), have shown great potential and have been used to treat various cancers as single therapies or in combination with other modalities. However, despite this remarkable progress, patients with TNBC have shown a low response rate to this approach, commonly developing resistance to immune checkpoint blockade, leading to treatment failure. Extracellular acidosis within the tumor microenvironment (also known as the Warburg effect) is one of the factors preventing immune cells from mounting effective responses and contributing to immunotherapy treatment failure. Therefore, reducing tumor acidity is important for increasing cancer immunotherapy effectiveness and this has yet to be realized in the TNBC environment. In this study, the oral administration of sodium bicarbonate (NaHCO_3_) enhanced the antitumor effect of anti-PD-L1 antibody treatment, as demonstrated by generated antitumor immunity, tumor growth inhibition and enhanced survival in 4T1-Luc breast cancer model. Here, we show that NaHCO_3_ increased extracellular pH (pH_e_) in tumor tissues in vivo, an effect that was accompanied by an increase in T cell infiltration, T cell activation and IFN-γ, IL2 and IL12p40 mRNA expression in tumor tissues, as well as an increase in T cell activation in tumor-draining lymph nodes. Interestingly, these changes were further enhanced in response to combined NaHCO_3_ + anti-PD-L1 therapy. In addition, the acidic extracellular conditions caused a significant increase in PD-L1 expression in vitro. Taken together, these results indicate that alkalizing therapy holds potential as a new tumor microenvironment immunomodulator and we hypothesize that NaHCO_3_ can enhance the antitumor effects of anti-PD-L1 breast cancer therapy. The combination of these treatments may have an exceptional impact on future TNBC immunotherapeutic approaches by providing a powerful personalized medicine paradigm. Therefore, our findings have a great translational potential for improving outcomes in TNBC patients.

## 1. Introduction

Triple-negative breast cancer (TNBC) is the most aggressive breast cancer (BC) subtype and accounts for 15–20% of all breast cancers [1]. Patients with TNBC have limited therapeutic options, such as chemotherapy, and a very poor prognosis [2]. Currently, there is no clinically validated, molecularly targeted therapy for these patients [3] and immunotherapies have been emerging as a new and promising option for TNBC treatment. Immunotherapeutic agents exert their antitumor effect by enabling the host immune system to target and eliminate tumors. The most explored of these therapies are monoclonal antibodies, which target cytotoxic lymphocyte antigen-4 (CTLA-4), and the programmed cell death protein 1 (PD-1)/programmed death-ligand 1 (PDL-1) axis. Expressed on cancer cells, PD-L1 mediates immune evasion by binding to the T cell PD-1 receptor, halting the T cell response [4]. Blocking the PD-L1/PD1 pathway using immune checkpoint blockers (ICBs) reactivates T cells. To date, several therapeutic monoclonal antibodies that target either PD-1 (nivolumab, pembrolizumab) or PD-L1 (avelumab, atezolizumab, durvalumab) have been FDA approved for treating various cancers such as melanoma, kidney, bladder, gastric, lung and head and neck cancers [5,6,7,8,9,10,11,12,13,14,15]. Unfortunately, in TNBC patients, ICBs as a single therapy have shown relatively low clinical response rates (15–20%) [16].

TNBC patients have been shown to have significantly greater PD-L1 expression, as well as higher numbers of tumor-infiltrating CD8+ lymphocytes, than those with other BC types [4,17]. These two markers have been associated with higher survival rates and better response rates to chemotherapy [17] and hence are considered as favorable prognostic markers in these patients. In addition, PD-L1 expression has been associated with the efficacy of anti-PD-1/PD-L1 treatments [18]. Therefore, targeting PD-1 or PD-L1 has a strong potential as a therapeutic strategy in TNBC [4,17]. For metastatic TNBC, anti-PD-L1 immune checkpoint inhibitor atezolizumab has recently been FDA approved in combination with chemotherapy for patients whose tissues test ≥1% positive for PD-L1 expression [16]. The Phase III Impassion 130 trial, evaluating atezolizumab in combination with nab-paclitaxel in patients with metastatic or locally advanced TNBC that expressed PD-L1, showed clinical overall survival benefits in patients treated with this combination of drugs [16]. Despite this remarkable progress, most advanced cancer patients still showed intrinsic or naturally acquired resistance to immune checkpoint blockade, leading to treatment failure.

One of the critical factors that often renders immunotherapy ineffective is extracellular acidosis within the local tumor microenvironment [19,20,21,22,23,24,25]. Poor tissue perfusion and reduced buffering capacity in the extracellular tumor microenvironment significantly contributes to a decrease in tumor extracellular pH (pHe), with tumor intracellular pH (pHi) remaining slightly more basic, as demonstrated by 31P MRS [26,27]. Several magnetic resonance imaging (MRI) methods have been developed to measure in vivo pH. MRI spectroscopies of endogenous inorganic phosphate or exogenous shift reagents have been used to measure pHe [28]. The pHi-pHe gradient in a variety of tumors vs. normal tissue, measured in vivo by 31P MRS, clearly demonstrated tumor pHe as acidic, with pHi ranging from neutral to alkaline [29,30]. In human TNBC tumors, pHe measured by microelectrode showed values in good agreement with the values observed in animal systems [31], i.e., acidic pHe ranging from 6.2 to 7.0 [31,32], while tumor pHi was either neutral or alkaline [33], and this pH gradient was not observed in normal tissues. In a preclinical immunodeficient MDA-MB-231 TNBC mouse tumor model, a non-invasive MRI method demonstrated acidic pHe [34,35]. In the immunocompetent 4T1 mouse model, pHe was also reported to be significantly acidic [36]. Acidic tumor pHe is a consequence of the high aerobic and glycolytic rate of tumor cells and a significant attribute of the tumor microenvironment (TME) [19,37]. Recent evidence suggests that acidic pH in TME affected the effector functions of T and NK cells, pro-tumor macrophage polarization and the immunosuppressive functions of myeloid-derived suppressor cells (MDSCs) [25,26]. Acidic pH induced infiltrating T cells and NK antitumor immune cells to enter an anergic, inactivated state and promoted the recruitment and activity of immunosuppressive pro-tumor MDSCs and regulatory T cells [24]. Acidosis was also demonstrated to boost exosome release from regulatory T cells (Tregs), further enhancing immunosuppression [25]. In vitro, acidic pH (6.5) suppressed T cell functions, including IL-2 secretion and activation of T cell receptors [30]. In conclusion, tumor microenvironment acidity was demonstrated to inhibit antitumor immune effectors and stimulate pro-tumor elements.

In addition to promoting tumor progression, invasion and metastasis, tumor acidosis also induces drug resistance [38,39,40]. It has been shown that oral administration of NaHCO_3_ increased intra-tumoral pH, which resulted in an inhibition of tumor growth and metastasis in murine models [23,24,25]. It has also been reported that tumor acidity may influence responses to immunotherapy [40,41,42]. Recent studies have shown that lactate induced an increase in PD-L1 expression in Lewis lung carcinoma and melanoma cells, while tumor growth was significantly reduced in response to treatment with both NaHCO_3_ and PD-L1 inhibitor [43,44,45,46]. Oral NaHCO_3_ administration was previously used to modulate the acidic pHe in solid tumors by several groups [32,36,47]. Raghunand et al. have shown that NaHCO_3_ effectively reversed pH gradients in tumors, but not in normal tissues [30]. Pilon-Thomas et al. reported that increasing tumor pHe with NaHCO_3_ increased response to immune checkpoint inhibitors, which was associated with increased T cell infiltration in a B16 melanoma murine model [45]. These proof-of-concept studies indicated the potential of acidic pHe within the tumor microenvironment as a therapeutic target for improving the responses to cancer immunotherapies. Reversed pH gradient in breast cancers can be restored by the systemic administration of NaHCO_3_ and this alkaline treatment might be an effective complementary cancer treatment [48] and a successful strategy to maximize the efficacy of ICBs [27].

In this study, we used a 4T1-Luc breast cancer mouse model to investigate the potential of the neutralization of extracellular acidic pH to enhance anti-PD-L1 cancer therapy. Our findings imply that, in the 4T1-Luc breast cancer mouse model, acidic pH_e_ led to the upregulation of inhibitory immune checkpoint molecules, in vitro and in vivo, and that the neutralization of tumor acidosis significantly increased the efficacy of anti-PDL1 treatment in reducing tumor growth and in increasing survival. In addition, pH neutralization markedly enhanced the inflammatory response as well as the antitumor immune response of tumor-infiltrating T cells compared to anti-PD-L1 treatment alone. Thus, our findings indicate that the oral administration of alkalizing agent may improve the efficacy of anti-PD-L1 immunotherapy by neutralizing the acidic tumor microenvironment. Our work has an exceptional potential to impact clinical cancer care, since all the components used in this work have already been individually used in the clinical setting.

## 2. Material and Methods

### 2.1. Antibodies

Fluorescent dye-conjugated antibodies specific to CD45, CD4, CD8, CD69, CD8, CD3, CD44, CD47 and IFN-γ and Streptavidin -APC, PD-L1 were obtained from Bio-legend (San Diego, CA, USA).

### 2.2. Cell Line

Murine 4T1 cells transduced with firefly luciferase (4T1-Luc) were purchased from American Type Culture Collection (ATCC) (Manassas, VA, USA). Cells were cultured in a complete RPMI-1640 medium (ThermoFisher Scientific, Waltham, MA, USA) supplemented with 8% fetal bovine serum and 1% penicillin/streptomycin and maintained at 37 °C with 5% CO_2_. The pH of the complete medium was adjusted to pH 6.5 and pH 7.4 with concentrated hydrochloric acid and sodium hydroxide, respectively.

### 2.3. 4T1 Breast Cancer Mouse Model and Treatment

We used 6- to 8-week-old female BALB/c mice (Jackson Laboratories). The mice were housed at the Henry Ford Hospital animal facilities (Detroit, MI, USA) under pathogen-free conditions. Animal protocols were approved by the Henry Ford Health IACUC committee and all the studies complied with ethical regulations and humane endpoints. The murine orthotropic 4T1 mouse tumor model was prepared as previously described [49]. In brief, 6-week-old syngeneic immunocompetent BALB/c female mice were inoculated with 4T1 cells (5 × 10^4^) into the mammary fat pad (MFP). We used 4T1-Luc cells that express luciferase to enable in situ visualization by bioluminescence imaging (BLI). The tumor-bearing mice were randomly divided into five treatment groups (n = 6) as follows: (1) control (water only), (2) glucose, (3) NaHCO_3_, (4) anti-PDL1, (5) NaHCO_3_ + anti-PD-L1. The mice in group number three were given drinking water with 200 mM NaHCO_3_ to increase the pHe > 7.2 via bicarbonate-induced metabolic alkalosis [45,48,50]. The mice in group number two were given drinking water with 5% D-glucose to lower the pHe ≤ 6.5 [51]. NaHCO_3_ or 5% of glucose treatment started three days prior to tumor injection and continued until the end of the experiment. The control mice received regular tap water. Tumor growth was monitored daily and measured every 3 days by digital calipers. Tumor volume was determined as length (mm) × width (mm) × height (mm) × 0.5. The tumors were allowed to grow for 8 days, at which point the mice in groups number four and five were treated intraperitoneally with 200 µg of αPD-L1 monoclonal antibody (clone: J43, Bio Cell) in 50 µL PBS at intervals of 3 to 4 days until the end of the experiment.

### 2.4. pH Measurement

The animals were sedated using isoflurane. Both the needle microelectrode and the reference electrode were obtained from Microelectrodes, Inc., Bedford, NH, USA (MI-401F and MI-408B, respectively, Microelectrodes). The mice were anesthetized and a shallow, small (5 mm) incision was made in an alternate (non-tumor) site where the 1 mm reference electrode was placed subcutaneously. A needle microelectrode was inserted up to 1.3 cm into the center of the tumor and held in place until the pH readings stabilized. The needle was rotated once in each location to allow the pH electrode to be reread at the same depth to make two independent measurements per location. The pH was measured at three locations: one near the center or core of the tumor, one in the mid-region of the tumor and one at the rim of the tumor; these values were averaged to a mean value for each animal. All the measurements were performed after calibration with pH 4.0, pH 7.0 and pH 10 reference solutions.

Urine was obtained by applying gentle abdominal pressure against the mouse for 10 to 20 s over aluminum foil. Urine was collected by micropipette and transferred to a tube for pH measurement.

### 2.5. In Vivo Bioluminescence Imaging (BLI)

Seven to ten minutes before the imaging procedure, the mice were treated with an intraperitoneal (ip) injection of the luciferin substrate (150 mg/kg). The mice were then anesthetized using isoflurane 1–3% in O_2_ for the induction phase, followed by 2% in O_2_ for the maintenance phase. Once anesthetized, the mice underwent BLI. Quantification of the in vivo luciferase signal of the tumor area was performed using Living Image software version 4.8 (Xenogen Corporation-Perkins Elmer, Hopkinton, MA, USA) by defining a photon emission region of interest (ROI) for each animal. The data are expressed as fold increase of photons/seconds/cm^2^/sr over time. A ROI signal equal to or more than 9 × 10^9^ photons/seconds/cm^2^/sr was considered as an indicator of tumor progression in the absence of any sign of suffering.

### 2.6. Flow Cytometry

Tumors, tumor-draining lymph nodes and spleens were harvested under sterile conditions. Single-cell suspensions were prepared, and red blood cells were removed using ACK lysis buffer for the tumor and spleen. Tumor cell suspensions were prepared from the solid tumors by enzymatic digestion in HBSS (Life Technologies, Carlsbad, CA, USA) using 1 mg/mL collagenase and 0.1 mg/mL DNases I (all from Sigma-Aldrich, Burlington, MA, USA) with constant stirring for 2 h at 37 °C. The resulting suspension was passed through a 70 μm cell strainer and washed once with a complete RPMI. The cells were then stained with fluorescently labeled antibodies as follows: fluorescein isothiocyanate-, phycoerythrin-, Cyanine7 or biotin-conjugated monoclonal antibodies for CD16/32 (Fc blocker) (BioLegend, San Diego, CA, USA 101302; 93), CD3 (145-2C11), CD4 (N318), CD8 (53-6.7), CD45 (53-7.3), CD69 (H1.2F3), CD44, CD47 and PD-L1. For the in vitro experiments, 4T1-Luc breast cancer cells were grown in RPMI-1640 containing 8% FBS and 1% penicillin/streptomycin at 37 °C with a medium pH between 6.5 and 7.4. Cells were analyzed by flow cytometry for cell surface PD-L1 expression and for intracellular IFN-γ expression using specific fluorescently labeled antibodies (Anti-PD-L1 from BioLegend and anti-IFN-γ-PE (XMG1.2) from BD Biosciences). Cells were analyzed using LSR II equipped with four lasers (BD Biosciences, Franklin Lakes, NJ, USA) and FlowJo software version 10.

### 2.7. Cytokines Gene Analysis by q-PCR

Total RNA was isolated using TRIzol reagent (Invitrogen, Carlsbad, CA, USA) following the manufacturer’s instructions. Isolated RNA was reverse transcribed into cDNA using synthesis kit (Bio-Rad, Hercules, CA, USA). Real-time RT-PCR was performed using iTaq Universal SYBR Green Supermix (Bio-Rad). Individual expression data were normalized to GAPDH. The primers used were as follows:IFNγ, 5′-TTCTTCAGCAACAGCAAGGC-3ʹ and 5ʹTCAGCAGCGACTCCTTTTCC-3′;IL-2, 5′-GCGGCATGTTCTGGATTTGACTC-3′ and 5′-CCACCACAGTTGCTGACTCATC-3′;IL-4, 5′-AACGAGGTCACAGGAGAAGG-3′ and 5′-TCTGCAGCTCCATGAGAACA-3′;IL-12, 5′-ACGAGAGTTGCCTGGCTACTAG-3′ and 5′-CCTCATAGATGCTACCAAGGCAC-3′;GAPDH, 5′-GCCAAGGTCATCCATGACAACT-3′ and 5′-GAGGGGCCATCCACAGTCTT-3′.

The mRNA levels of the test genes were normalized to the expression of GAPDH.

### 2.8. Statistics

Each bar graph represents the mean ± standard deviation. Significance was calculated using Student’s *t* test. A *p* value of less than 0.05 was considered statistically significant.

## 3. Results

### 3.1. Extracellular Acidosis Increases PD-L1 Expression in 4T1-Luc Breast Cancer Cells

The tumor microenvironment is characterized by acidic conditions with a pH_e_ level between pH 6.2 and pH 6.9 [52]. To determine the relationship between extracellular acidosis and PD-L1 expression in breast cancer cells, we acidified the pH of the cell culture medium using HCl. Cell surface PD-L1 expression was significantly increased under the acidic pH, as shown by flow cytometry (Figure 1A,B). In addition, PD-L1 mRNA expression in the 4T1-Luc breast cancer cells was also upregulated under these acidic conditions (Figure 1C and Appendix A).

### 3.2. Sodium Bicarbonate Therapy Decreases Tumor PD-L1 Expression In Vivo

To determine the effect of pH_e_ on PD-L1 expression in the tumor microenvironment in vivo, we treated 4T1-Luc tumor-bearing mice with NaHCO_3_ solution by oral gavage. The animals receiving NaHCO_3_ treatment exhibited a decrease in tumor-associated PD-L1 expression compared to the non-treated animals. This decrease was observed at the PD-L1 protein and mRNA levels (Figure 2A–C). Orally administered NaHCO_3_ reduced PD-L1 expression through pH neutralization of the acidic tumor microenvironment.

### 3.3. Tumor Growth Inhibition and Survival

We evaluated the potential of NaHCO_3_ to enhance the antitumor effects of anti-PD-L1 therapy in the 4T1-Luc mouse model by measuring tumor growth. The studies were conducted using different pH conditions in vivo. The mice were divided into five experimental groups that were treated with tap water, glucose, NaHCO_3_, anti-PD-L1 antibodies and the combination of NaHCO_3_ and anti-PD-L1 antibody. All the mice exhibited rapid primary tumor growth (Figure 3A–C). The tumor volume increased at the fastest rate in the glucose-treated mice, reaching maximal tumor volume at Day 23. The NaHCO_3_ treatment group exhibited similar tumor growth to that of the control group. Treatment with anti-PD-L1 was more effective compared to NaHCO_3_ and glucose alone; however, no significance was observed, indicating that anti-PD-L1 immunotherapy alone had a moderate anticancer effect. However, when anti-PD-L1 therapy was combined with NaHCO_3_ therapy, tumor volumes were significantly reduced compared to each monotherapy and the controls, indicating that NaHCO_3_ could improve primary tumor responses to anti-PD-L1 (Figure 3B). In addition, in response to the combination regimen, tumor weights were significantly lower compared to all the other treatment groups (Figure 3C). We also used BLI imaging to assess tumor growth by weekly whole-body BLI over a period of 22 days. The animals treated with the combination of NaHCO_3_ and anti-PD-L1 treatments exhibited significant tumor growth delay compared to the control group (Figure 3D). Next, we determined the effect of combined NaHCO_3_ and anti-PD-L1 treatment on survival. The mice treated with NaHCO_3_ alone or in combination with anti-PD-L1 exhibited an increase in survival compared to the untreated mice or the mice treated with glucose alone, but no significance was observed (Figure 3E).

To evaluate the changes in the extracellular pH in response to the therapy with NaHCO_3_, we used a pH electrode to assess the extracellular pH (Figure 3F). Consistent with previous reports, the untreated animals exhibited an acidified extracellular pH of 6.8 or above [45,53]. However, the mice treated with glucose exhibited an acidified extracellular tumor environment with a pH of 6.6 or below. On the other hand, oral administration of NaHCO_3_ significantly increased the extracellular pH to a neutral level of pH 7.3 or above. In numerous previous studies, this effect was shown to be specifically localized within the tumors, without causing any systemic alkalization [48]. Since the acidity of the TME may contribute to the observed resistance to anti-PD-L1 antibody treatment, increasing the pH_e_ of the TME may increase the antitumor effectiveness of the anti-PD-L1 therapy and may be used as a novel therapeutic approach.

### 3.4. Changes in Urine pH and Body Weight in Response to Treatment

To confirm that NaHCO_3_ does not cause any adverse effects, we measured urinary pH and animal body weight over the course of the experiment. In the mice receiving the oral NaHCO_3_ treatment, urine pH increased over the course of 22 days, as compared to urine pH in the control mice; however, this increase was within a physiological pH range (Figure 4A). A similar increase was also observed in the mice treated with the combination of NaHCO_3_ and anti-PD-L1 antibody. However, the anti-PD-L1 treatment alone did not cause an increase in urine pH, which had similar values to those detected in the control mice. In addition, there were no significant changes (*p* values greater than 0.05) in body weight in any of the experimental groups on Day 8, 14 or 22 after treatment (Figure 4B). These results indicate that neither the NaHCO_3_ treatment nor the combination of NaHCO_3_ and anti-PD-L1 antibody treatments had any significant adverse effects.

### 3.5. Immune Cells Population in Tumor Tissues, Tumor-Draining Lymph Nodes (TDLN) and Spleen

Tumor-infiltrating lymphocytes are crucial effectors in antitumor immune response that can significantly improve the efficacy of cancer therapy. To investigate changes in the tumor-associated T cell population in response to the NaHCO_3_ and anti-PD-L1 treatment combination, we generated a single cell suspension from collected tumor tissues and analyzed it by flow cytometry. Single treatments with either NaHCO_3_ or anti-PD-L1 significantly increased the frequency of CD8+ T cells within the tumors compared to the control and glucose-treated animals. More importantly, treatment with NaHCO_3_ enhanced the anti-PD-L1-induced CD8+ T cell tumor infiltration (Figure 5A,C). However, the frequency of the CD4+ T cells did not increase in response either to NaHCO_3_ alone or to anti-PD-L1 alone (Figure 5B,C).

We next hypothesized that combination therapy might facilitate the trafficking of TDLNs and spleen CD8+ and CD4+ T cells. To test this hypothesis, a single cell suspension was prepared from collected auxiliary, brachial and inguinal TDLNs and spleens and analyzed by flow cytometry. In the TDLNs, NaHCO_3_ or the combination of NaHCO_3_ and anti-PD-L1 treatments resulted in a decrease in the frequencies of CD8+ T cells (Figure 5D). At the same time, these treatments did not elicit any changes in the frequencies of CD4+ T cells (Figure 5E,F). In contrast, the CD8+ and CD4+ T cell populations within the spleen were not affected by any of the treatments (Figure 5G,H).

Next, we evaluated whether NaHCO_3_ enhanced anti-PD-L1-induced T cell activation in vivo by analyzing the frequencies of CD69+ T cells in the collected tumor tissues, TDLNs and spleens. Flow cytometry analyses of the tumor-associated CD3+ T lymphocytes showed significant increase in CD69 expression in response to the combined NaHCO_3_ and anti-PD-L1 treatment. Interestingly, a significant increase in the frequency of CD69+ T cells was also detected in response to the NaHCO_3_ treatment alone, while the anti-PD-L1 treatment alone did not have any effect on these cells (Figure 6A). These data indicate that anti-PD-L1 treatment alone is insufficient for T cell activation and that NaHCO_3_ may be responsible for abolishing the inhibitory mechanisms involved. Similar changes were observed in the TDLNs; however, T cell activation by the NaHCO_3_ treatment alone did not reach significance compared to the controls (Figure 6B). In the spleen, no changes in the frequencies of CD69+ T cells in response to any of the treatments were observed (Figure 6C). Representative flow cytometry dot plots analyzing CD8+ T cell activation are shown in Figure 6E. We also tested the expression of IFN-γ in tumor-associated CD8+ T cells. The frequencies of tumor-associated CD8+ T cells expressing IFN-γ were increased in response to NaHCO_3_ alone or the combination of NaHCO_3_ and anti-PD-L1 treatments (Figure 6D). These results indicate that the combination therapy may enhance cytotoxic T cell responses and that NaHCO_3_ may be used to improve cancer patients’ sensitivity to immunotherapies via the IFN-γ pathway mechanisms [54].

### 3.6. Combination of NaHCO_3_ and Anti-PD-L1 Treatments Augments the Expression of Inflammatory Cytokines in TNBC Tumor Tissue

We investigated the expression of IFN-γ, IL-2, IL-4 and IL-12 inflammatory cytokines by q-PCR within the tumor tissues. The expression of IFN-γ, IL-2 and IL-12 mRNA was significantly increased in response to NaHCO_3_ alone and the combination of NaHCO_3_ and anti-PD-L1 treatments compared to the control group (Figure 7A,B,D). In the case of IL-4, there was no significant difference in the mRNA expression levels when compared between the groups (Figure 7C).

### 3.7. CD47 and CD44 Expression in 4T1-Luc Tumor Cells In Vivo

We also evaluated the cell surface expression of CD47 and CD44 molecules in response to the NaHCO_3_ and anti-PD-L1 treatments, in a single cell suspension generated from the 4T1-Luc tumor tissues. A significant change was observed only in the expression of the CD47 molecule, which was downregulated in response to the combination of NaHCO_3_ and anti-PD-L1 treatments (Figure 8A). No change was observed in the CD44 expression in response to any of the treatments (Figure 8B).

## 4. Discussion

Acidic pHe may promote the escape of solid tumors from immune surveillance and potentially limit the efficacy of immunotherapies such as anti-PD-L1 therapy [45,55]. Numerous studies have recently elucidated the mechanisms of resistance to PD-L1 immunotherapy [56,57,58]. It is reported that tumor acidity can influence responses to immunotherapy [40,41]. Orally delivered bicarbonate has been shown to effectively reverse acidity in tumors [53,59]. In several mouse models, alkalization of the tumor milieu inhibited invasive tumor growth and the formation of metastases [45,53,60,61,62]. However, in aggressively growing cancers like the B16 melanoma, no tumor inhibition was observed [45,53] and pH neutralization had a beneficial effect only when combined with immune checkpoint inhibitors or adoptive cell transfer [45]. These studies indicate the importance of understanding the differences in tumor microenvironments between different cancer types in exploiting the effects of cancer acidification.

In the current work, we demonstrate that treatment with an alkalinizing NaHCO_3_ agent enhances the therapeutic effect of anti-PD-L1 immunotherapy in a 4T1-Luc breast cancer mouse model. These increased responses were most likely a result of an alkalinizing agent increasing the tumor pHe. PD-L1 expression in different TNBC cell lines have also been reported [63]. Culturing 4T1-Luc cells in low pH conditions resulted in higher expression of PD-L1 mRNA and surface. These results are in agreement with previous studies showing elevated level of PD-L1 expression in the presence of acidic conditions compared with basic conditions [64]. We also found that NaHCO_3_ treatment alone reduced primary tumor growth compared to the control treatments, which agrees with previous studies showing that bicarbonate monotherapy significantly inhibited tumor growth rates in vivo [47]. However, our results with NaHCO_3_ treatment did not reach significance, which can be explained by the cancer cell type dependent variability in response to bicarbonate treatment alone.

In this study, we show that the combination therapy of NaHCO_3_ alkaline reagent and immune checkpoint inhibitor anti-PD-L1 effectively suppressed tumor growth in 4T1-Luc TNBC mouse model. The progression and development of breast cancer showed a positive correlation with acidosis. Previously, acidic pH_e_ was demonstrated to play a role in contributing to tumor growth [65] and inhibiting certain immune functions that are important for the success of immune checkpoint therapies [66].

Recent studies have shown that tumor-infiltrating lymphocytes are closely related to the prognosis and outcomes of breast cancer patients. Our findings reveal that NaHCO_3_ or the combination of NaHCO_3_ with anti-PD-L1 therapy lead to a significant increase in CD8+ and CD4+ T cell infiltration in tumor tissues. We also evaluated CD8+ T cells and CD4+ T cells in TDLNs and the spleen. In response to NaHCO_3_ or the combination of NaHCO_3_ and anti-PD-L1 treatments, the frequencies of CD8+ T cells significantly decreased in the TDLNs, while there was no change in in the frequencies of CD4+ T cells. Previous reports have shown that TDLNs play a pivotal role in initiating antitumor immunity through processes that involve tumor antigens processing by antigen-presenting cells (APCs) and APCs’ migration to TDLNs, where they present antigens and initiate a primary immune response [67]. Another surprising finding was that of tumor-infiltrating lymphocyte activation and IFN-γ secretion in response to treatment conditions. We showed that NaHCO_3_ alone or the combination of NaHCO_3_ and anti-PD-L1 treatments significantly increased T cell activation and IFN-γ production from cell-specific CD8+ T cells. IFN-γ has a prominent role in immune checkpoint blockades with anti-PD1 or anti-PD-L1. IFN-γ was found to be localized to regions of high PD-L1 expression and T cell infiltrates in tumor, and these tumor-associated CD8+ T cell can further inhibit the antitumor immune response through an IFN-γ-driven increase in PD-L1 expression [68]. This mechanism of adaptive immune resistance may explain tumor escape from immunosurveillance. On the other hand, the increased levels of IFN-γ also indicate the activation of a cytotoxic T cell response, which has also been reported to possibly lose its function at acidic pH [19,45,69,70]. IFN-γ also drives the upregulation of PD-L1 expression, and its production by T cells is required to mediate their therapeutic effect [46,71,72]. In addition to IFN-γ production, acidic pH_e_ can stimulate increases in cell surface PD-L1 expression, as we demonstrated here in the 4T1-Luc breast cancer cell line, in vitro. It is probable that under in vivo conditions this mechanism may contribute to the overall increase in PD-L1 expression in the tumor microenvironment and the inhibition of the antitumor immune response. In addition to IFN-*γ*, the increase in IL2 and IL12 in response to NaHCO_3_ or the combination of NaHCO_3_ and anti-PD-L1 treatments indicated the induction of specific antitumor responses. Lastly, we found that the combination therapy downregulated CD47 surface molecules within the TME. Cell surface CD47 was shown to bind to SIRP alpha protein on the macrophage surface, which initiates a “don’t eat me” signal, an important contributor to cancer cell immune evasion [31]. Altogether, our findings indicate that the addition of NaHCO_3_ to anti-PD-L1 therapy enhances antitumor immunity.

## 5. Conclusions

Here, we demonstrate that NaHCO_3_ therapy is an effective adjuvant for anti-PD-L1 treatment. We found that the combination of NaHCO_3_ and anti-PD-L1 markedly triggered tumor suppression in TNBC tumor-bearing mice. In addition, we found that the NaHCO_3_-enhanced therapeutic efficiency of PD-L1 was accompanied by the induction of an effector CD8+ T cell response and the increased accumulation of IFN-γ + CD8+ T cells within the tumor. This research demonstrates the enormous potential of NaHCO_3_ therapy as a modifier of the TNBC tumor microenvironment and as an adjuvant to enhance PD-L1 treatment activity.

## Figures and Tables

**Figure 1 cancers-15-04931-f001:**
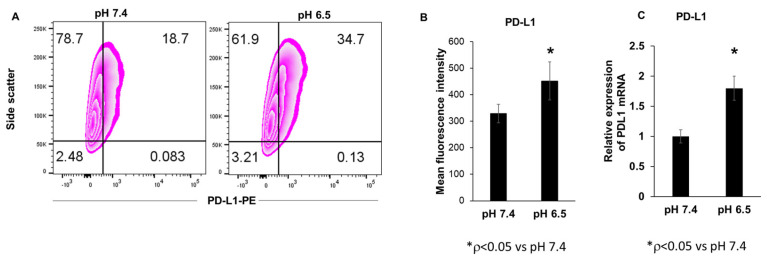
Effect of extracellular acidosis on PD-L1 expression in 4T1-Luc cells. The pH of the cell culture medium was adjusted by HCl or NaOH, and 4T1-Luc cells were incubated for 24 h and analyzed by flow cytometry. Flow cytometry dot plots (**A**) and mean fluorescence (**B**) of cell surface PD-L1 expression (* *p* < 0.05 for pH 6.5 vs. pH 7.4) and (**C**) PD-L1 mRNA levels analyzed by qPCR (* *p* < 0.05 for pH 6.5 vs. pH 7.4).

**Figure 2 cancers-15-04931-f002:**
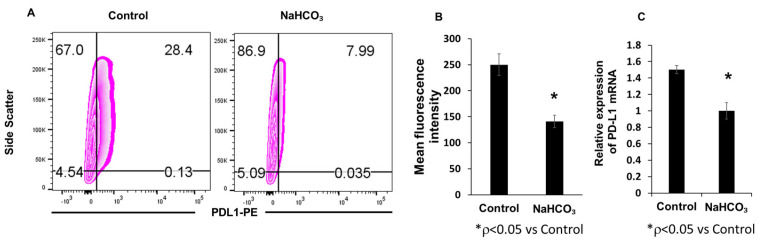
Effect of NaHCO_3_ on PD-L1 expression in vivo. Mice were treated with 200 mmol/L sodium bicarbonate solution or regular tap water (control) by free oral ingestion from Day 3 to Day 23. Flow cytometry dot plots (**A**) and mean fluorescence (**B**) of tumor tissue PD-L1 expression (* *p* < 0.05 for NaHCO_3_ vs. control). (**C**) Tumor tissue PD-L1 mRNA analyzed by qPCR (* *p* < 0.05 for NaHCO_3_ vs. control).

**Figure 3 cancers-15-04931-f003:**
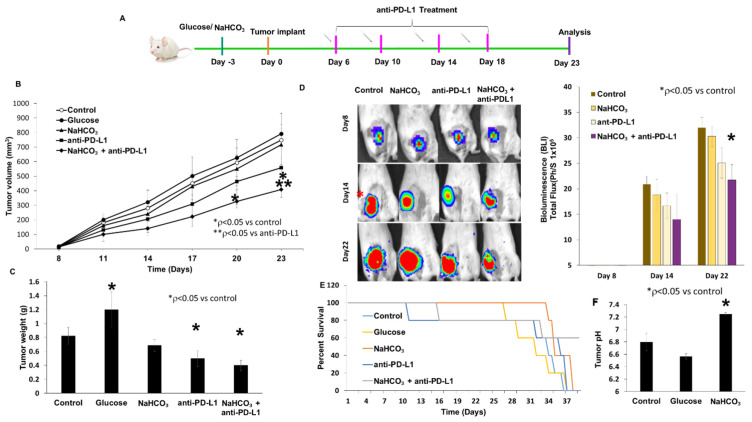
Effect of combined anti-PD-L1 plus NaHCO_3_ therapy on tumor growth, intratumor pH and survival of mice. (**A**) Experimental timeline of 4T1-Luc tumor inoculation and anti-PD-L1 antibody treatment. (**B**) Tumor volumes were measured every 3 days and at the endpoint (* *p* < 0.05 for NaHCO_3_ + anti-PD-L1 vs. control; ** *p* < 0.05 for NaHCO_3_ + anti-PD-L1 vs. anti-PD-L1). (**C**) Resected tumor weights at the endpoint (* *p* < 0.05 for glucose, NaHCO_3_ + anti-PD-L1 vs. control). (**D**) Primary tumor growth monitored weekly using BLI. Total flux of the tumor region was quantified at each corresponding time point (photon/sec/cm^2^) (* *p* < 0.05 for NaHCO_3_ + anti-PD-L1 vs. control). (**E**) Survival curve of mice with treatments and followed until moribund. (**F**) Intratumor pH measured by microelectrode in response to NaHCO_3_ or glucose treatment (* *p* < 0.05 for NaHCO_3_ vs. control).

**Figure 4 cancers-15-04931-f004:**
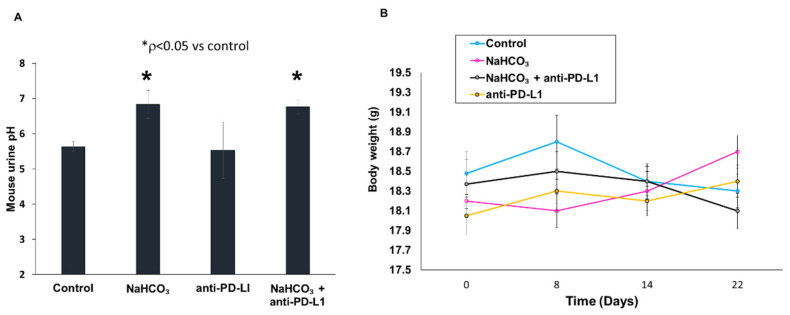
Effect of NaHCO_3_ on urine pH and body weight. (**A**) Urine pH in response to NaHCO_3,_ anti-PD-L1 and the combination of NaHCO_3_ and anti-PD-L1 treatments, Day 23 (* *p* < 0.05 vs. control). (**B**) Body weight change in response to NaHCO_3_, anti-PD-L1, and NaHCO_3_ and anti-PD-L1 treatments (* *p* < 0.05 vs. control).

**Figure 5 cancers-15-04931-f005:**
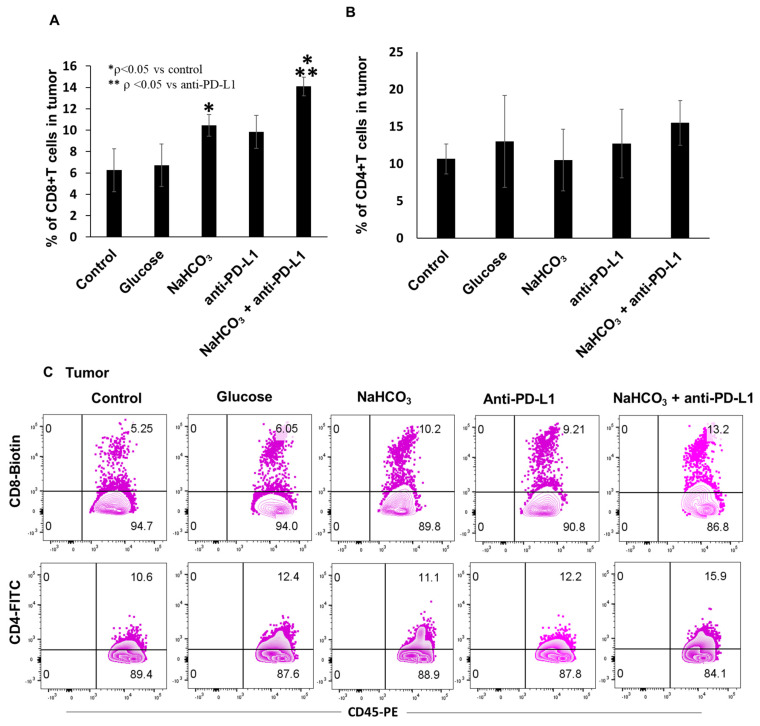
Frequencies of immune CD8+ and CD4+ cells in response to treatment in tumors, TDLNs and spleen (**A**) The frequencies of cytotoxic CD8+ T cells by flow cytometry in tumor (* *p* < 0.05 for NaHCO_3,_ NaHCO_3_ + anti-PD-L1 vs. control; ** *p* < 0.05 for NaHCO_3_ + anti-PD-L1 vs. anti-PD-L1). (**B**) The frequencies of CD4 helper T cells by flow cytometry in tumor. (**C**) Representative flow cytometry dot plots of intratumor CD8+ and CD4+ cell frequencies. The frequencies of CD8+ T cells (**D**) and CD4+ T cells (**E**) in tumor-draining lymph nodes. Frequencies of CD8+ T cells (**G**) and CD4+ T cells (**H**) in spleen. (**F**) Representative flow cytometry dot plots of CD8+ and CD4+ cells in lymph nodes.

**Figure 6 cancers-15-04931-f006:**
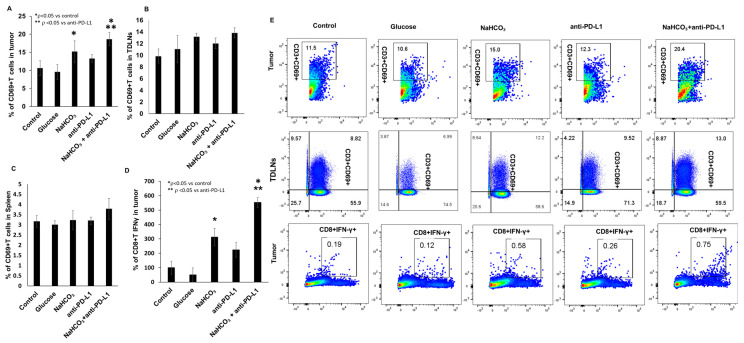
CD69 expression in CD3+ T cells in tumor, TDLNs and spleen. Frequencies of CD69+ T cells in tumors (**A**) TDLNs (**B**) and spleen (**C**). Cell-specific CD8+ T cells IFN-γ production measured by flow cytometry (**D**) (* *p* < 0.05 for NaHCO_3_, NaHCO_3_ + anti-PD-L1 vs. control; ** *p* < 0.05 for NaHCO_3_ + anti-PD-L1 vs. Anti-PD-L1). The flow cytometric cell population analyzed and shown as dot plot (**E**).

**Figure 7 cancers-15-04931-f007:**
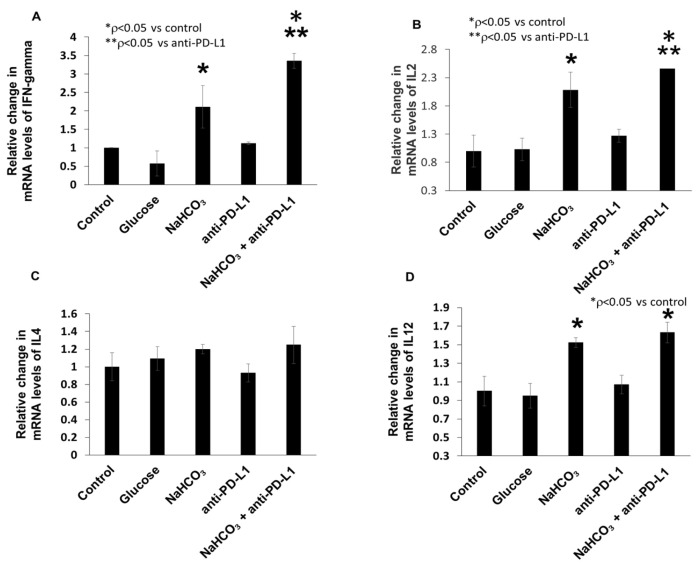
Levels of IFN-γ, IL2, IL12 and IL4 mRNA in tumor tissues. (**A**) The expression of IFN-γ mRNA level in tumor (* *p* < 0.05 for NaHCO_3_, NaHCO_3_ + anti-PD-L1 vs. control; ** *p* < 0.05 for NaHCO_3_ + anti-PD-L1 vs. ant-PD-L1). (**B**) The expression of IL2 mRNA level in tumor (* *p* < 0.05 for NaHCO_3_, NaHCO_3_ + anti-PD-L1 vs. control; ** *p* < 0.05 for NaHCO_3_ + anti-PD-L1 vs. anti-PD-L1). (**C**) The expression of IL4 mRNA level in tumor. (**D**) The expression of IL12 mRNA level in tumor tissue (* *p* < 0.05 for NaHCO_3_, NaHCO_3_ + anti-PD-L1 vs. control). Expression levels are normalized to the expression of GAPDH housekeeping gene.

**Figure 8 cancers-15-04931-f008:**
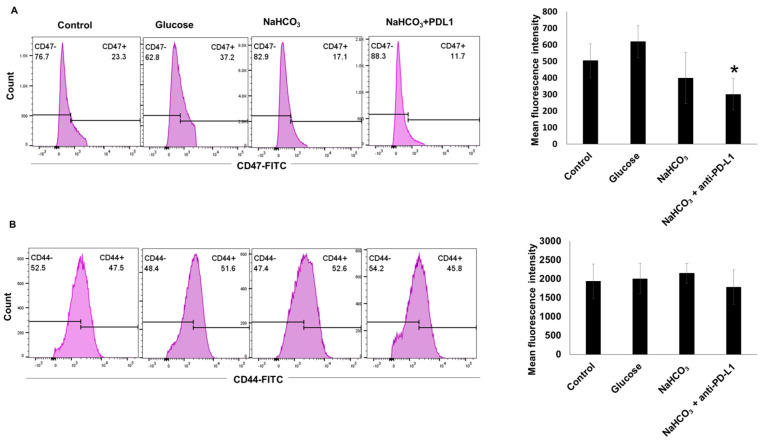
CD47 and CD44 expression in tumors. (**A**) Flow cytometry histograms showing the percentage of CD47+ cell population and mean fluorescence intensity (* *p* < 0.05 for NaHCO_3_ + anti-PD-L1 vs. control). (**B**) Flow cytometry histograms showing the percentage of the CD44+ cell population and mean fluorescence intensity.

## Data Availability

The research data used in this study can be shared upon reasonable request sent to the corresponding authors.

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
