# Peer review of "Immunotherapy Enhancement by Targeting Extracellular Tumor pH in Triple-Negative Breast Cancer Mouse Model"

_cancers, 2023, doi:10.3390/cancers15204931_

Round 1
Reviewer 1 Report
In the presented study on immunotherapy enhancement by targeting extracellular tumor pH in a TNBC mouse model, the increase in extracellular pH accompanied by T cell infiltration, T cell activation, and the expression of inflammatory cytokines is interesting. While the presented data in this manuscript is good, the authors should consider the following major issues to improve the quality of this manuscript
1. Why did the authors employ 4T1 Luciferase cells throughout the entire study? It is advisable to include 4T1 parental cells, particularly for in vitro studies, to ensure a comprehensive investigation.
2. To demonstrate the cell-specific nature of pH changes and PDL1 expression, consider examining the expression of PDL1 across different TNBC cell lines
3. Repeated administration of the PDL1 antibody is known to induce hypersensitivity reactions and result in cell death in mice. Have you observed such reactions, and if so, how did you address it?
4. Some of the references were not cited correctly; for example, reference 49 before Figure 3 was not cited appropriately. Please arrange the references carefully.
5. To enhance the robustness of the data, include statistics as individual data points in the results wherever possible
6. There are some words with missing spaces in between them. Please correct them, for example, check the last paragraph in the discussion.
Author Response
Please, find below uploaded file as response to reviewer.

Reviewer 2 Report
General comments
- Since the authors did not use the journal templete , It is difficult to write the comments specifically.
- The title "Immunotherapy Enhancement by Targeting Extracellular Tumor pH in Triple Negative Breast Cancer Mouse Model" is very interesting to add more insights for researchers and cilinicians how to enhance immunotherapy in breast cancer as well as others.
- Suggest to do protein analysis to confirm whether the addition of NaHCO3 have a significant effect on the protein levels of selected markers in this study.
specific comments
- There is no any statistical evidence to say " there were no significant changes in body weight in any of the experimental groups over the course of experiment (Figure 4B)". Suggest to put the P-value
- Expression of IFN-g, IL-2 and IL-12 mRNA were significantly increased in response to NaHCO3 alone and the combination of NaHCO3 and anti-PD-L1 treatment, compared to the control group (Figure 7A, 7B, 7D).On figure 7D does not show the change in combinations, suggest to focus on the changes either alone or in combination of NaHCO3 with Anti-PDL1.
Author Response

(The authors gave the same response as above.)

Round 2
Reviewer 1 Report
The authors have corrected the manuscript, and I hope the revised version is of a quality that can be accepted for publication